# Thermal Reading of Texts Buried in Historical Bookbindings

**DOI:** 10.3390/s24175493

**Published:** 2024-08-24

**Authors:** Stefano Paoloni, Giovanni Caruso, Noemi Orazi, Ugo Zammit, Fulvio Mercuri

**Affiliations:** 1Department of Industrial Engineering, University of Rome Tor Vergata, 00133 Rome, Italy; stefano.paoloni@uniroma2.it (S.P.); zammit@uniroma2.it (U.Z.); mercuri@uniroma2.it (F.M.); 2Institute of Heritage Science, National Research Council of Italy (ISPC-CNR), Montelibretti, 00010 Rome, Italy; giovanni.caruso@cnr.it

**Keywords:** infrared thermography, optically semi-transparent materials, ancient books, buried text detection, readability

## Abstract

In the manufacture of ancient books, it was quite common to insert written scraps belonging to earlier library material into bookbindings. For scholars like codicologists and paleographers, it is extremely important to have the possibility of reading the text lying on such scraps without dismantling the book. In this regard, in this paper, we report on the detection of these texts by means of infrared (IR) pulsed thermography (PT), which, in recent years, has been specifically proven to be an effective tool for the investigation of Cultural Heritage. In particular, we present a quantitative analysis based, for the first time, on PT images obtained from books of historical relevance preserved at the Biblioteca Angelica in Rome. The analysis has been carried out by means of a theoretical model for the PT signal, which makes use of two image parameters, namely, the distortion and the contrast, related to the IR readability of the buried texts. As shown in this paper, the good agreement between the experimental data obtained in the historical books and the theoretical analysis proved that the capability of the adopted PT method could be fruitfully applied, in real case studies, to the detection of buried texts and to the quantitative characterization of the parameters affecting their thermal readability.

## 1. Introduction

In the field of Cultural Heritage (CH), non-destructive evaluation techniques [1,2,3] are increasingly used since they allow researchers to gather relevant information about the investigated items, such as that concerning their structures or the processes adopted in their manufacture. In this respect, it is worth noting that most of the valuable information is quite often obtained from features located beneath the sample surface that cannot be probed by means of ordinary optical inspection techniques. Consequently, in recent years, there has been considerable research work devoted to the development of experimental techniques for the investigation of features buried inside artworks, which, among other techniques, has led to the establishment of active Infrared Thermography (IRT) as a very effective tool for this kind of study [4,5,6,7]. The IRT working principle relies on heating the sample, typically through the absorption of visible (VIS) light, and on the subsequent locally resolved detection of the induced variation in the sample’s infrared (IR) emission by means of an IR camera [8,9]. As in the present study, the pulsed IRT (PT) configuration, which makes use of VIS light pulses delivered by flash lamps to heat the sample, has become the preferred one for CH surveys because of it is simple and fast to use. In this case, the IR camera is employed to record a sequence of images, which, in the following, are referred to as thermograms, describing the map of the local thermal state of the sample at different delay times from the onset of the heating pulse. One of the peculiar abilities of PT is given by the possibility of distinguishing between features located at different depths into the sample, which is not possible with most of the alternative techniques employed in the CH field, such as IR reflectography. In this respect, it is worth noting that thermography enables the detection of subsurface features, provided that these are located within the diffusion length of the induced temperature rise. In fact, the presence of local subsurface inhomogeneities may induce significant modifications in the corresponding heat diffusion rate, leading to a different local temperature and therefore IR emission at the surface area above the inhomogeneity with respect to the surroundings and creating a thermal contrast in the recorded thermographic images. Therefore, the analysis of the thermograms obtained while varying the temperature diffusion length may allow us to distinguish features located at different depths in the sample. Such variation can be obtained by either varying the modulation frequency in the case of periodic heating, such as in the lock-in thermography (LIT), or allowing the heat to diffuse for longer times in the case of the step heating or the PT configurations, hence probing larger depths into the sample. Concerning the investigations shown in the present manuscript, LIT has also been proved to be an effective tool for the detection of buried texts [10]. However, to date, a theoretical model for the quantitative analysis of LIT experimental results concerning buried graphical features has not yet been developed. In addition, as mentioned previously, depth-resolved LIT investigations require different measurements to be performed while varying the heating modulation frequency. In this respect, PT may be considered an extremely valid alternative since it enables thermographic inspections to be carried out faster with respect to LIT.

In order to exploit PT’s ability to carry out quantitative depth-resolved studies in artworks, over the last several years, research efforts have been devoted to the development of theoretical models for the analysis of the time dependence of the PT signal. According to such models, the relation between the PT signal and the induced temperature distribution in the sample may depend strongly on the optical properties of the sample. For optically opaque samples, such as those made of metals like the bronze statues, both the VIS light absorption and the IR emission occur at the sample surface; therefore, the PT signal is merely proportional to the temperature variation at the sample surface [5]. Consequently, the time dependence of the IR emission is solely influenced by the sample’s thermal properties, since the optical ones only affect the PT signal amplitude [5]. Conversely, in artifacts made of semi-transparent materials, such as books, both the VIS light absorption and the IR emission processes occur over the sample volume; therefore, the sample’s optical and thermal properties both play a crucial role in establishing the time dependence of the PT signal [10]. Consequently, the modeling of the PT signal is less straightforward than in opaque samples since it arises from the interplay between optical propagation and thermal diffusion processes [11,12].

The present manuscript concerns a particular category of CH, that of book heritage, mostly constituted by semi-transparent materials. In the literature, many studies, based on the use of numerous techniques, have been dedicated to the characterization of library artefacts and, in particular, to the recovery of lost writings, like in the case of erased and overwritten texts in palimpsests [13]. In these applications, ultraviolet illumination [14], hyperspectral imaging [15] and multispectral imaging [16,17] have been used for the recovery of the damaged or censored texts, for the study of burned papyri and parchments and for the reading of erased inks on parchment [14], respectively. Of particular importance among these studies are those concerning the reading of texts buried in the bookbindings of historical manuscripts [10]. In fact, since the 16th century, because of the increase in book production associated with the invention of printing, it became very common to use earlier book material for the production of new bookbindings [18]. In particular, reused written fragments were applied to support the structure between the board and the spine or inserted between the cover and the endpapers to strengthen the connection between the binding and the book block [19]. However, these interesting fragments are often not accessible by a simple visual inspection, because they are mostly hidden between the book cover and the endpapers. It is therefore of utmost importance to have available a technique that enables the characterization of these fragments without dismantling the bookbinding, in order to reveal valuable information for the dating and provenance of the manuscript. In addition, these fragments may come from books that are more valuable than the book that hosts them, or they may contain unpublished historical content.

This paper deals with the PT investigation of texts buried in two original ancient books preserved at the Biblioteca Angelica of Rome and the capacity of PT to read them. In the following, after briefly describing the main aspects affecting the readability issue and reviewing the peculiarity of the PT signal originating from buried ink elements, we present the experimental results obtained in texts lying at different depths in historical bookbindings. In particular, we report on the quantitative analysis of the contrast and the distortion, two parameters that affect the quality of the thermographic images of the detected texts and, therefore, their readability. It is indeed the application of this quantitative thermographic approach to the readability analysis of historical buried texts that represents the main novelty of this work. 

## 2. Thermal Reading of Buried Texts: Signal Theory

### 2.1. Thermography and Texts Buried in Historical Bookbinding

Recently, some studies have been presented where PT was successfully employed to investigate graphic elements lying beneath paper and parchment layers [10,11,12,20]. However, these studies are based on the analysis of specially made laboratory samples and do not include the validation of the proposed method on original manuscripts. On the other hand, PT has been successfully used for the study of historical bookbindings [21,22] with written or printed scraps inserted between the cover and the endleaf [18,19]. In this respect, PT has proved to be very effective in detecting the texts lying on such scraps that otherwise could no longer be accessed [10]. In addition, many studies carried out on historical artifacts have shown that thermographic imaging allows their thermal reading, an extremely important result for codicologists and paleographers. These studies have clearly shown that the possibility of obtaining images of a hidden text that are sharp enough to be read depends on some structural characteristics that have not yet been quantitatively analyzed in historical artifacts. In particular, they include the nature of the overlying materials and of the interface between such materials and the text support and, above all, the depth of the text with respect to the outer surface.

As an example, Figure 1 shows two thermograms retrieved from recorded sequences and processed in a way described later on in Section 3.2. They display buried texts located at different depths in the same bookbinding. In particular, in the thermogram recorded at the earlier delay time (see Figure 1b), the written parts buried at a depth of 95 μm become visible, while in that obtained at a later delay time (see Figure 1c), the text buried at a depth of 155 μm can also become evident, partly overlapping the shallower text but also extending laterally beyond its position. It can be seen that the text shown in Figure 1c does not possess the same degree of readability as that shown in Figure 1b because of its larger depth. In fact, as discussed in more detail later on in Section 4, under similar experimental conditions, the features shown by the recorded thermograms become less contrasted and more distorted with the increase in their depth, the distortion being affected by the lateral heat diffusion. Actually, besides the decrease in the thermogram’s contrast, it was found that the text readability was also negatively affected by lateral heat diffusion effects. In fact, even when employing the uniform VIS light illumination of the sample surface, when the light absorption is not homogeneous, the generated heat diffuses according to a 3D regime [12]. 

This situation corresponds to the case of the studied buried texts, since a more efficient heat generation occur at the ink position and then diffuses into the less absorbing colder surroundings along both the directions perpendicular and parallel to the sample surface. Therefore, the apparent width of the ink strokes that can be retrieved from the recorded thermograms increases with the time delay while its sharpness decreases, leading to the blurring of the PT images. This is confirmed in the thermograms obtained on the same buried text at different values of time delay, shown in Figure 2, where the edges of the ink characters become progressively smeared out because of the progressive increase in lateral heat diffusion with time. 

Concerning the study of the buried texts, it is worth mentioning that also alternative techniques employing scanning X-ray fluorescence (XRF) [23] or computed tomography [24,25] have been quite recently proposed. The XRF technique even enables the elemental composition evaluation of even the fragments located deep into the bookbinding. However, the technique proves to be effective only for the investigation of texts written with iron-based inks and, unlike PT, cannot be applied to the case where carbon-based inks are used. If must be consider that, in order to visualize the buried scraps, it is often necessary to dismantle the books that may be of historical relevance. It is therefore of paramount importance to optimize both the experimental configuration and the data processing method of the employed techniques to improve the possibility of reading such buried texts. To this aim, a number of models for the PT signal originating from ink features buried in between adjacent paper layers have recently been presented by the authors of this work [11,12]. Such models employed, according to the analyzed situation, either the 1D or the 3D heat diffusion regimes and they were then tested on specifically designed laboratory samples. In particular, the results in Ref. [10] have confirmed the capability of PT to detect texts buried beneath book endpapers. In Ref. [11], the same authors provided a model for PT signal analysis considering a one-dimensional heat diffusion regime for the analysis of subsurface ink features. In Ref. [12], the model was extended to account for 3D heat diffusion, particularly for the analysis of the lateral diffusion effects affecting the quality of the thermal reading of finite size inked elements. Unlike the previously mentioned studies which were based on the analysis of ad hoc prepared laboratory samples, the present paper deals with PT applied to real case studies and, in particular, to the analysis of the readability of texts buried in historical bookbindings. In the following, after briefly reviewing the main aspects related to the PT signal originating from buried graphical features, we report experimental PT results that have been obtained from buried texts located at different depths of the studied historical bookbindings. Thereafter, we present the quantitative analysis of both the contrast and the distortions of the recorded PT images, which are then related to the physical and structural properties of the investigated samples.

### 2.2. PT Theory

As mentioned above, PT may enable the detection of buried ink features even if their presence induces only marginal changes in the induced temperature distribution in the investigated item. In fact, unlike the case of optically opaque materials, in semi-transparent media, the contrast in the IR emission may originate from inhomogeneities in the optical properties as well as the thermal ones, such as the VIS light absorption and IR emissivity of the buried ink layers are most often larger than those of the surrounding medial. 

Based on the considerations reported above, theoretical models for the PT signal originating from ink features buried in a semi-transparent material have been developed and described in detail elsewhere [11,12]. Briefly, according to the schematic representation depicted in Figure 3, in such PT signal models, a uniform sample slab of thickness *H* is considered, where a thin graphical element is located at depth *d*. As a result of the VIS light absorption taking place both within the sample slab and at the buried ink feature, a time-varying 3D temperature distribution $T\left(\bar{x},z,\;t\right)\;$is induced, where $\bar{x}\;$and *z* are directions parallel and orthogonal to the sample surface, respectively (see Figure 3). 

Concerning the PT signal $S\left(\bar{x},t\right)$, the IR radiation emitted from the sample volume must be considered; therefore, the following expression for the signal can be obtained [11]:(1)$$S\left(\bar{x},t\right)=K\left[\int_{0}^{d}\beta T\left(\bar{x},z,t\right)e^{-\beta z}dz+\eta T\left(\bar{x},t\right)e^{-\beta d}+\int_{d}^{H}\left(1-\eta \right)\beta T\left(\bar{x},z,t\right)e^{-\beta z}dz\right]\;$$
where *K* is a constant accounting for different experimental factors, such as the VIS light intensity, the IR radiation detection efficiency and other geometrical aspects; *β* is the effective value of the material’s IR absorption coefficient over the detected wavelength range; and *η* is the ink layer’s IR absorptance. It is worth noting that the expression for the signal originating from ink-layer-free areas is equivalent to only the first integral evaluated between 0 and *H*.

In Figure 4, possible PT signal profiles $S\left(\bar{x},t\right)$ over the edge of a buried ink feature are shown. According to the 1D diffusion regime, where only the heat diffusion along the direction orthogonal to the sample surface is considered, one would predict a sharp change in the signal profile, as sketched by the dotted line, because of the greater IR radiation emission by the buried drawing with respect to that by the weaker-absorbing surroundings. However, as the ink layer gets heated by the VIS light transmitted through the overlying paper layer and by the temperature field diffusing from the surroundings, the heat also starts diffusing from the ink toward the surrounding cooler regions along the $\bar{x}\;$direction, thus leading to a smoothing of the temperature profile over the ink edge, as outlined by the gray plot in Figure 4. In order to account for the characteristics of the PT signal profile over the ink edges, two specific parameters of the recorded thermograms have been introduced. With reference to the notation adopted in Figure 4, the first one is the contrast $C=S_{max}-S_{min}$, which is defined as the difference between the PT signal values detected over the buried ink feature and where no ink is present, respectively. It is clear that the text readability is ensured, among other factors, by large enough values of *C*. The second parameter is the distortion index, ∆, corresponding to the distance between the values of $\bar{x}\;$where the signal differs by a detectable fraction (2%) from its maximum and minimum measured values:$\;{∆=x}_{max}-x_{min}$ [12] (see Figure 4). Basically, the distortion index measures how much the detected $S\left(\bar{x},t\right)\;$profile differs from the step-like one, where ∆ = 0.

In Figure 5, the time dependence of both the contrast and the distortion index following the onset of the heating pulse are shown. In these simulations, both the optical and thermal properties of the paper layer have been assumed to be equal ones typically found in the literature for paper (*α* = 1.3 × 10^2^ cm^−1^; *β* = 2.4 × 10^2^ cm^−1^; *D* = 1.0 × 10^−3^ cm^2^/s) [11], while those of the ink were set equal to those of paper except for the absorptance set at *η* =1. As can be seen in Figure 5a, the contrast reaches its maximum value after a typical delay time that increases with the ink depth. The decrease in the maximum value of the contrast with the ink depth is associated with a larger attenuation of both the heating VIS light and of the temperature field reaching the layer. An additional factor is the larger attenuation of the IR radiation emitted by the buried graphical feature during the propagation through the paper over-layer before it emerges from the paper surface. The shift of the maximum occurs because of the time required by the heat to diffuse from the shallow volume where the VIS light is absorbed to the ink layer position where the IR emissivity is largest. At later times, a decrease in *C*(*t*) can be observed because of both the ongoing heat diffusion past the ink layer position and along the lateral direction, as well as because of the heat losses from the sample surface to the surroundings. Concerning the distortion index (see Figure 5b) it turns out to show the same profile for all the ink depths, displaying a progressive increase depending only on the delay time. This is due to the fact that the index is solely dependent on the time during which lateral diffusion occurs, a process governed by the value of the thermal diffusivity of the material, while the ink depth only affects the image contrast. On the basis of the plots reported in Figure 5, it may be gathered that the effective readability of a given subsurface graphical feature depends on its position. In fact, with the increase in depth, the maximum contrast is both decreased in amplitude and shifted to larger delay times, with a consequent increase in the overall lateral diffusion and, therefore, in the distortion in the recorded thermogram.

## 3. Samples and Experimental Setup

### 3.1. Samples

Thermographic investigations were performed on two original ancient books preserved at the Biblioteca Angelica of Rome. The first one is a book printed in 1758 (ʘ.2.16) with a board parchment binding and uncolored endpapers (Figure 1a). Its binding is characterized by having two superimposed spine wings sandwiched in between the endpapers and the boards. The wings were made from scraps of reused paper, both written on one side. The cross-section of this structure is sketched in Figure 6a. The second book was printed in 1592 (f.9.31). It has a limp parchment binding with uncolored endpapers (Figure 2) attached to written spine wings with the text at the outermost surface (Figure 6b).

### 3.2. Experimental Setup

The measurements were carried out by means of the PT setup hereafter described. Sample heating was obtained by means of two flash lamps of 3 kW maximum electrical power, oriented at 45° with respect to the sample surface, delivering pulses a few ms long. The used flash lamps allow the light power to be adjusted to achieve an adequate signal-to-noise ratio without exposing the artifact to excessive amounts of radiation. In addition, their light pulses were short enough to allow the application of the adopted theoretical model for the analysis of the experimental data, which implied a very short light pulse. The emitted light from the lamps was filtered in order to remove the IR spectral component and, consequently, to suppress spurious contributions to the PT signal originating from reflections of IR radiation at the sample surface. The detection of the IR radiation emitted by the sample was obtained using a Cedip JADE MWIR camera (320 × 240 pixels, InSb focal plane array, 30 μm pitch, 3.6–5.1 μm wavelength range), which is characterized by a Noise Equivalent Temperature Difference (NETD) < 25 mK at 30 °C. The Altair 5.50 software was used to process the IR images, which were recorded with a 150 Hz frame rate. Thanks to the above-mentioned specifications, the employed IR camera provided a sufficient spatial resolution and frame rate to obtain spatial and time sequences with an adequate amount of experimental data and a sensitivity in the operating spectral range high enough to ensure an adequate signal-to-noise ratio. In order to improve the contrast of the recorded thermograms, the frame obtained just before the heating pulse was subtracted from all subsequent frames constituting the recorded sequence.

## 4. Results and Discussion

Figure 7a–c show the thermograms obtained for increasing delay times (0.02 s, 0.05 s and 0.30 s) with respect to the heating light pulse, of the text I (see Figure 6) buried just beneath the endpaper in area I framed in blue, previously shown in Figure 1b. Figure 7d reports the PT signal profiles obtained from the three thermograms over the ink edge of a selected letter. The contrast of the thermographic signal shows similar values for the first two delay times, followed by a subsequent decrease with further delay, while the range of the distortion region (delimited by the vertical bars), corresponding to the distortion index value, increases continuously. Similarly, in Figure 8, the thermograms and the corresponding PT signal profiles, obtained at the same values of delay times used for the data in Figure 7, correspond to the area of a letter of the deeper text previously shown in Figure 1c, corresponding to area II framed in blue in Figure 1a (text II in Figure 6a).

Given the increased depth of its position with respect to that displayed in Figure 7, the contrast of the text is very small at early delay times; it then reaches a maximum and then decreases with increasing delay times, with the maximum occurring at later times with respect to that in Figure 7. The distortion index increases with the delay time in the same way as in Figure 7.

Figure 9 reports the corresponding results obtained over a letter of the text displayed in area III framed in black in Figure 2. The values of the contrast do not show substantial changes with increasing delay, while the distortion index correspondingly increases. Concerning the depth at which the three different analyzed texts are buried, the following considerations should be made. In the case of volume ʘ.2.16, the depths of texts I and II are unknown; one can only observe that text II is located at a depth larger than that of text I, as witnessed by the longer time required for text II to appear in the corresponding thermographic sequence. This is consistently sketched in the cross-section of Figure 6a, where the depth of text II is considered to be that of text I plus the thickness of the shallower wing placed over the deeper wing. Regarding text III of volume f.9.31, the presence of some tears and gaps in the end leaf allowed the determination of its thickness yielding a value of 105 ± 5 μm.

This thickness should, nevertheless, be considered only as approximate, as the effective thickness may be affected by the presence of the glue layer and possibly also by the mechanical effects caused by the pressure exerted during the gluing process. The effective depths of the three analyzed texts were determined as the values of the effective overlaying paper thickness that would yield the best agreement of the theoretical dependence of the contrast and distortion from the delay time with the corresponding experimental data as shown later on in Figure 10. The effective values turned out to be 95 μm, 110 μm and 155 μm, respectively, for texts I, II and III.

Finally, in Figure 10a,b, we summarize the significant information gathered from the results discussed so far concerning the influence of the depth of the investigated texts and the corresponding delay times for best thermographic readability. In particular, in Figure 10, we report the experimental results obtained from the previous graphs (open symbols) displayed on the corresponding theoretical (solid lines) and experimental (solid symbols) curves of the delay-time dependence of both the contrast and distortion indexes obtained from the thermogram sequences of the three analyzed buried texts.

The continuous lines were obtained by means of Equation (1) using values for the optical and thermal parameters typical of paper (previously referred to in the theory section) and effective paper thickness values that would yield the best agreement with the experimental data. The solid symbols in Figure 10a correspond to the differences between the PT signals recorded over the letter analyzed in the plots in Figure 7, Figure 8 and Figure 9 and a non-inked reference area. The following conclusions can be drawn:The excellent agreement between the theoretical values and the experimental data confirms the soundness of the proposed theoretical model for the PT signal. It also supports its capability to provide reliable indications of the values of the depth of the buried ink layer and of the parameters affecting the readability of the buried texts, once the values of the optical and thermal parameters of the involved paper layers are known, as well as the parameters affecting the readability of the buried texts.With the increasing depth of the buried ink layer, the peak contrast value decreases, and its occurrence is shifted to longer delay times.The time dependence of the distortion index shows that it is related only to the delay time of the detection of a particular subsurface text, irrespective of its depth, as confirmed by the superposition of the values corresponding to texts buried at different depths of the theoretical and experimental values obtained from Figure 7, Figure 8 and Figure 9.

Therefore, concerning the readability of the different texts, the following can be concluded:For the sufficiently shallow text (95 μm deep), the best readability is obtained in the earliest thermograms obtained at the earliest delay times (Figure 7a) because of the highest achieved contrast and smallest distortion.For the deepest text (155 μm), the largest contrast can be obtained at longer delay times (Figure 8b,c), thus compromising the possibility of optimal readability because of the corresponding increase in distortion.

## 5. Conclusions

In the present study, pulsed thermography was applied to the detection of text buried below the endleaves of ancient manuscripts of significant historical relevance. A quantitative analysis of the obtained thermographic results was carried out by using a theoretical model to elaborate on the quantities actually affecting the readability of the text retrieved in the thermograms at different delay times with respect to the pulsed heating source.

In particular, thermogram sequences (as a function of the delay time) of the texts buried at different depths below the endleaves of different ancient books housed in the Biblioteca Angelica in Rome were recorded. Two parameters were introduced, the contrast index and the distortion index, and they were used to evaluate the degree of readability of the texts retrieved in the thermograms. The contrast dependence on the delay time showed a peak feature, whose value decreased with the increasing depth of the buried ink layer and moved to larger delay times. Moreover, the distortion index was shown to be related only to the delay time of the detection of a particular subsurface text, irrespective of its depth. These results were confirmed both by theoretical calculations and by experimental results, with excellent agreement between them.

The depth of the subsurface features has been found to play a crucial role in the determination of the retrieved texts’ quality. For relatively shallow texts, the best readability is obtained in thermograms recorded at early delay times because of the sufficiently high achievable contrast and small distortion. For deeper texts, sufficient contrast can only be obtained at longer delay times, producing a larger distortion and therefore compromising the possibility of optimal readability.

Based on the consistency of the experimental method and the reliability of the proposed theoretical model, which closely reflected the results obtained experimentally, the presented approach makes pulsed thermography a very reliable method for the study of texts buried beneath paper sheets within the bookbinding of ancient books.

On the basis of the results reported in this work, in the near future, research efforts should be specifically devoted to the design of algorithms for the processing of PT images in order to improve the readability of the detected hidden texts, including procedures involving artificial intelligence. Moreover, further research activity should also be dedicated to the development of integrated approaches where PT is employed in combination with other possible imaging techniques in an effort to obtain complementary characterizations of the subsurface graphical features.

## Figures and Tables

**Figure 1 sensors-24-05493-f001:**
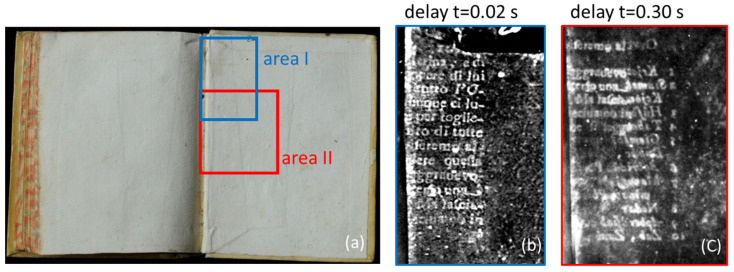
A book printed in 1758 (ʘ.2.16) from the Biblioteca Angelica of Rome: (**a**) a picture of the back endleaf; (**b**) a thermogram recorded 0.02 s after the light pulse, showing the text buried at a depth of 95 μm; (**c**) a thermogram recorded 0.30 s after the light pulse, also showing the text buried at a depth of 155 μm.

**Figure 2 sensors-24-05493-f002:**
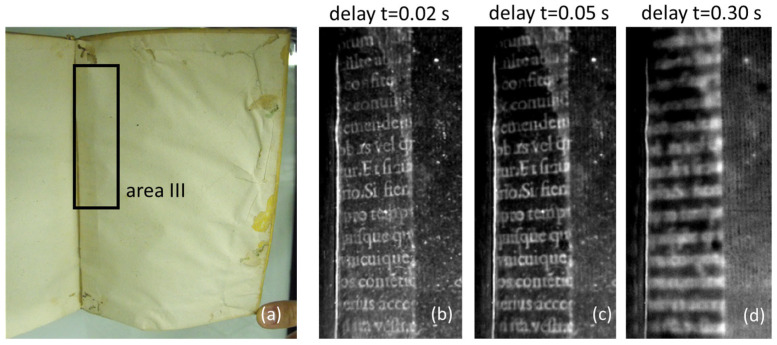
A book printed in 1592 (f.9.31) from the Biblioteca Angelica of Rome: (**a**) a picture of the back endleaf; thermograms of the black framed part (area III) recorded 0.02 s (**b**), 0.05 s (**c**) and 0.30 s (**d**) after the light pulse, showing the text buried at a depth of 110 μm.

**Figure 3 sensors-24-05493-f003:**
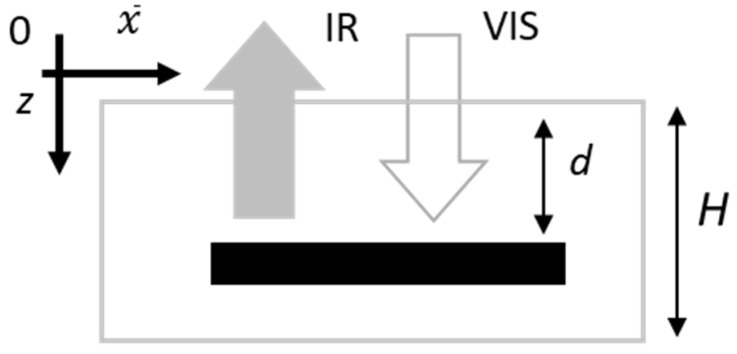
A sketch of the specimen considered in the model.

**Figure 4 sensors-24-05493-f004:**
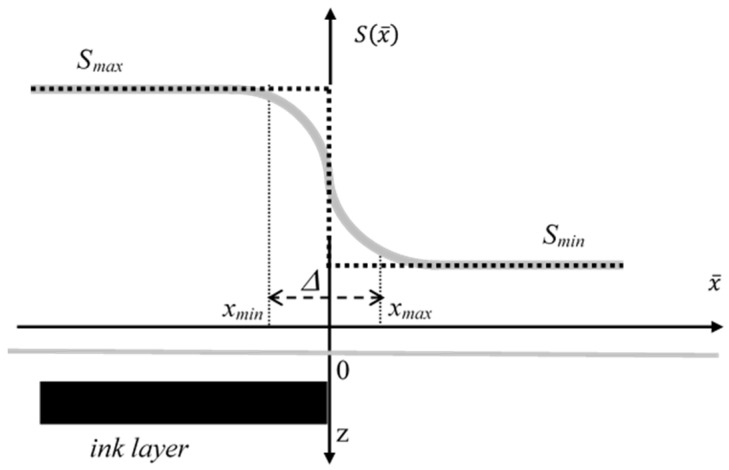
A sketch of the PT signal profiles over the edge at $\bar{x}$ = 0 of a subsurface ink feature, where 1D (black dotted line) and 3D (continuous gray line) heat diffusion regimes are considered. Also represented is the distortion index ∆ (see text).

**Figure 5 sensors-24-05493-f005:**
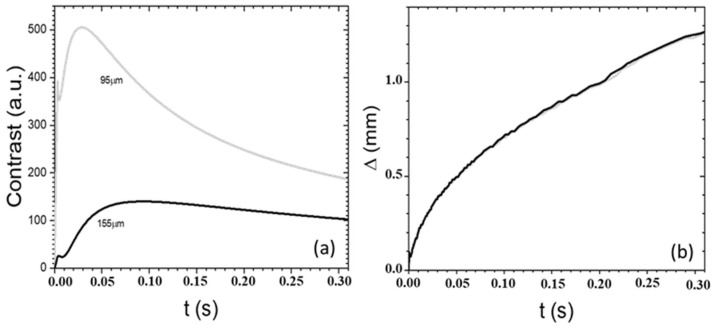
The theoretical delay-time dependence of the contrast *C*(*t*) (**a**) and the distortion index ∆(*t*) (**b**) over the edge of graphical features buried at different depths in a paper layer.

**Figure 6 sensors-24-05493-f006:**
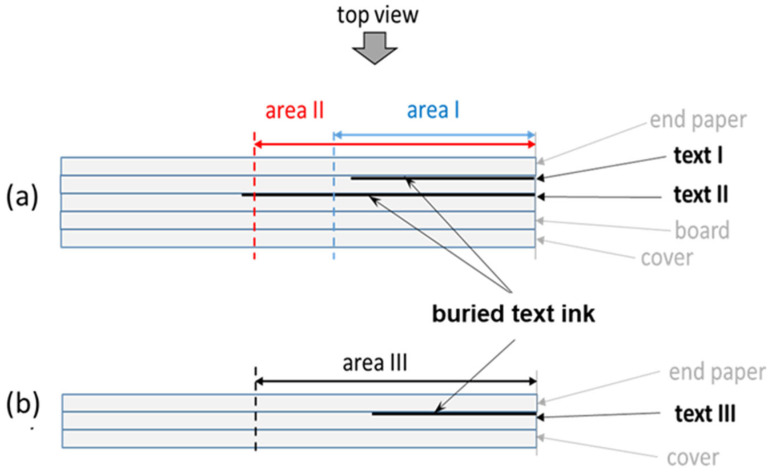
Sketches of the bookbinding cross-sections of (**a**) book ʘ.2.16 and (**b**) book f.9.31.

**Figure 7 sensors-24-05493-f007:**
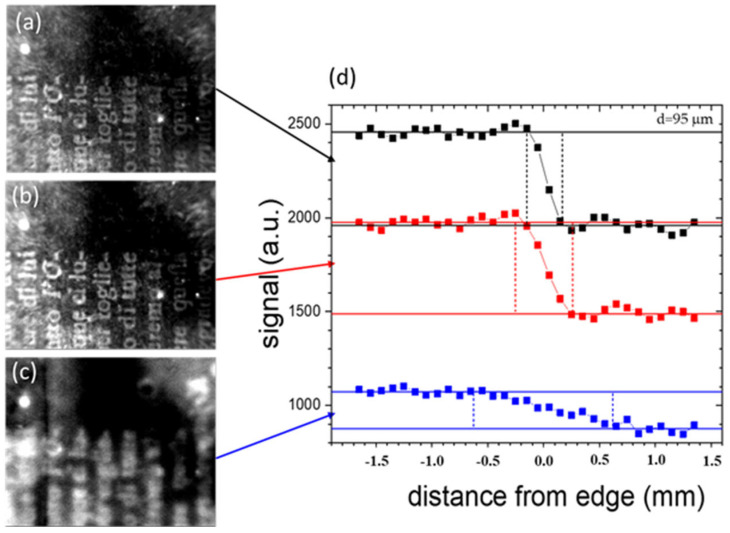
Thermograms of text I buried just beneath the endpaper in the area framed in blue previously shown in Figure 1b, obtained for increasing delay times of 0.02 s (**a**), 0.05 s (**b**) and 0.30 s (**c**) after the heating light pulse. (**d**) The PT signal profiles obtained over one of the letters.

**Figure 8 sensors-24-05493-f008:**
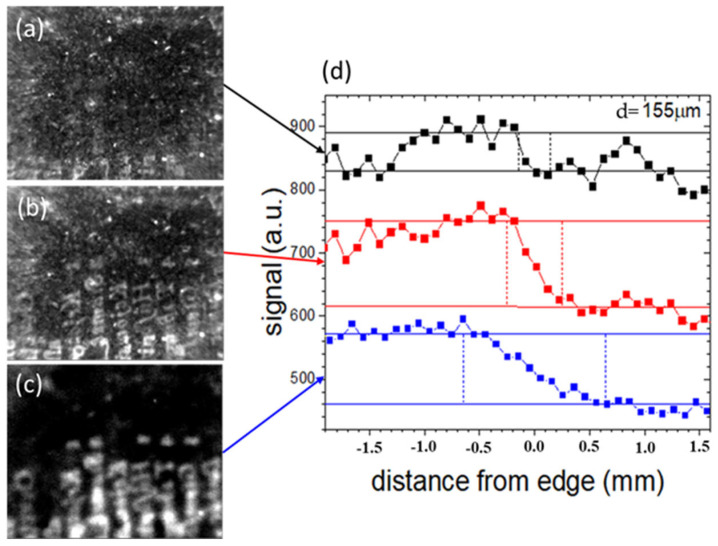
Thermograms of text II buried just beneath the endpaper in the area framed in red (previously shown in Figure 1c) obtained for increasing delay times of 0.02 s (**a**), 0.05 s (**b**) and 0.30 s (**c**) after the heating light pulse. (**d**) The PT signal profiles obtained over one of the letters.

**Figure 9 sensors-24-05493-f009:**
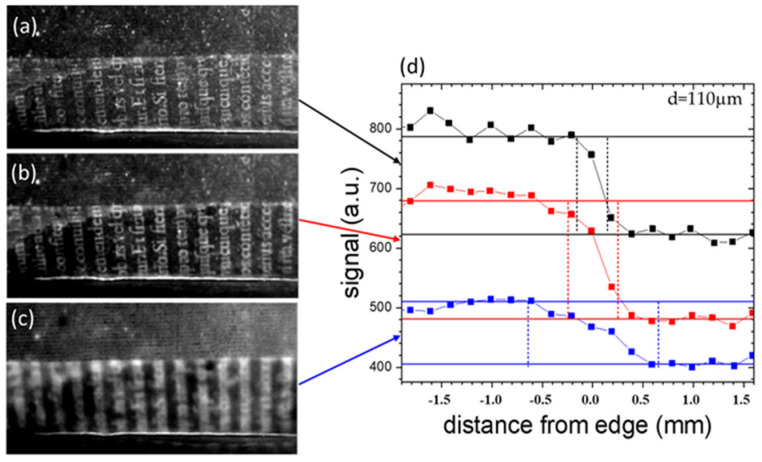
Thermograms of text III buried just beneath the area framed in black of the endpaper previously shown in Figure 2a, obtained for increasing delay times of 0.02 s (**a**), 0.05 s (**b**) and 0.30 s (**c**) after the heating light pulse. (**d**) The PT signal profiles obtained over one of the letters.

**Figure 10 sensors-24-05493-f010:**
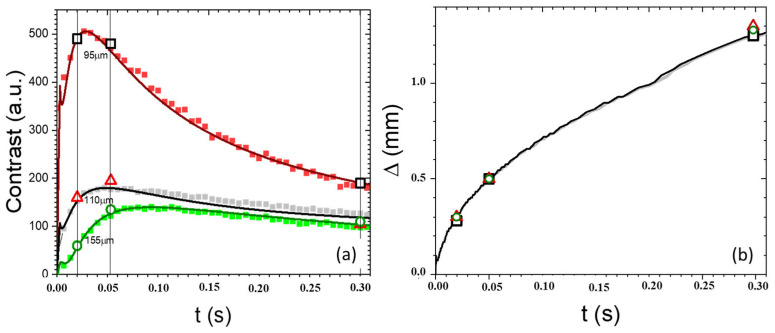
The time dependence of (**a**) the contrast *C*(*t*) and (**b**) distortion ∆ of the texts buried at different depths. The continuous lines represent the theoretical prediction, while the symbols correspond to the experimental data obtained according to the procedure described in the text.

## Data Availability

Further inquiries can be directed to the corresponding author.

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
