# Peer review of "Thermal Reading of Texts Buried in Historical Bookbindings"

_sensors, 2024, doi:10.3390/s24175493_

Round 1

Reviewer 1 Report

Comments and Suggestions for Authors

This paper pulsed thermography (PT)  has been specifically proven 13 to be an effective tool for the investigation of Cultural Heritage. 

1) Have you considered the lock-in thermography?

2) Have you considered  the thermography methods using other excitation? like eddy current, microwave, or laser? these methods are popular in NDT field.

3) the results are good. More comparision will be more useful.

thanks

Author Response

 1) Have you considered the lock-in thermography?

Yes, we have tried it outside the framework of this work [10] and in this regard it is important to point out that no theoretical model has yet been developed and tested for this specific application (optically transparent media), that would allow quantitative assessments of the depth of the hidden texts and of the aspects affecting their readability.

On the other hand, concerning the pulsed thermography, the rapidity and simplicity of the image acquisition and processing in this type of application, its effectiveness in providing a good readability of the buried texts and its capability to analyze the depth of the buried texts has been extensively proved.

The aim of the work is to show such capabilities in real applications where the pulsed thermography seems to be the most effective thermographic method. However, part of the above remarks concerning the lock-in thermography are now reported in the introduction (lines 55-66).

2) Have you considered the thermography methods using other excitation? like eddy current, microwave, or laser? these methods are popular in NDT field.

As mentioned above, the aim of the work was to show the capabilities and explore the limitations of the pulsed thermography in this specific application concerning the readability of the hidden text. Flash-lamps have been used as heating sources as they are the most readily available, least expensive, least complex heating sources providing spatially extended illumination at power levels safe for cultural heritage artworks.

3) The results are good. More comparison will be more useful.

In spite of this being only the first report concerning on field applications of buried texts recovery in ancient books, the variety of cases presented in the present paper in terms of different depths of the texts and also the case of overlapping texts, and together with the results previously presented on test samples [11-12,20] have proven the reliability and effectiveness of the proposed pulsed thermographic method for the non-destructive retrieval of texts  buried within the bookbinding of ancient books.

Reviewer 2 Report

Comments and Suggestions for Authors

This paper explores the use of infrared pulsed thermography (PT) to detect and read

texts hidden within the bookbindings of historical books, without dismantling them. The

study focuses on books preserved at the Biblioteca Angelica in Rome. A quantitative

analysis is presented, utilizing a theoretical model for the PT signal and employing

image parameters like distortion and contrast to assess IR readability. The results

demonstrate a strong correlation between experimental data and theoretical analysis,

validating PT’s effectiveness for revealing and characterizing buried texts in historical

manuscripts, the paper is recommended to be published only after the questions below are properly answered.

1. Introduction: The explanation of lateral heat diffusion and its effect on text

readability is clear. Provide a more detailed background on why this specific

study is important for historical manuscripts.

2. Page 2: The transition from discussing uniform VIS light illumination to heat

diffusion could be smoother. Providing a brief overview of previous methods and

their limitations.

3. Page 3: The description of Figure 1 should provide a clearer explanation of how

these thermograms were analyzed to determine text readability. Including more

context about the significance of the different depths (95 µm, 155 µm).

4. Page 6: Specify the criteria for choosing the particular flash lamps and IR

camera.

5. Page 7-10: Highlighting the key findings with bullet points or a summary table.

The conclusion is too brief and fails to adequately summarize the key findings

and their implications. It should also address potential future research directions

and applications, which are currently missing.

6.      There have been several papers to describe the application of infrared (IR) pulsed thermography on cultural heritage ,for example   Could you please illustrate the research innovation compared to them.

Mercuri, F., Orazi, N., Paoloni, S., Cicero, C., & Zammit, U. (2017). Pulsed thermography applied to the study of cultural heritage. Applied Sciences7(10), 1010.

Comments on the Quality of English Language

Quality of English Language is acceptable. 

Author Response

  1. Introduction: The explanation of lateral heat diffusion and its effect on text readability is clear. Provide a more detailed background on why this specific study is important for historical manuscripts.

A more detailed background of the studies devoted to the use of techniques for the recovery of readability of texts in historical manuscripts has been added. Introduction, pages 2-3, lines 84-103: “In the literature many studies, based on the use of numerous techniques, have been dedicated to the characterization of library materials and, in particular, to the recovery of lost writings, such as erased and overwritten texts in palimpsests [13]. In these applications, ultraviolet illumination [14], hyperspectral [15] and multispectral imaging [16-17] have been applied to the recovery of damaged or censored texts, to the study of burnt papyri and parchments, and to the reading of erased inks on parchment [14], respectively. Of particular importance among these studies are those concerning the reading of texts buried in the bookbindings of historical manuscripts [10]. In fact, from the 16th century onwards, it became very common to use earlier book materials for the production of new bookbindings, due to the increase in book production caused by the invention of printing [18]. In particular, reused written fragments were applied to support the structure between the board and the spine or inserted between the cover and endpapers to strengthen the connection between the binding and the book block [19]. However, these interesting fragments are often not accessible by the simple visual inspection, because they are mostly hidden between the book cover and endpapers. It is therefore of utmost importance to have a technique like that enables the characterization of these fragments revealing valuable information for the dating and provenance of the manuscript without dismantling its bookbinding”.

  1. Page 2: The transition from discussing uniform VIS light illumination to heat diffusion could be smoother. Providing a brief overview of previous methods and their limitations.

A brief overview of previous methods has been added (page 2, lines 47-66).

  1. Page 3: The description of Figure 1 should provide a clearer explanation of how these thermograms were analyzed to determine text readability. Including more context about the significance of the different depths (95 µm, 155 µm).

The thermograms in Fig. 1 are not intended to show the best readability of the two superimposed texts. Their purpose in this part of the work is only to show how the readability of the texts is strongly influenced by the depth of the text. The larger the depth of the text, the longer the delay needed to detect it and, consequently, the lower the contrast and sharpness of the text features. An explanation of how these thermograms were analyzed and compared with others of their sequence, in order to determine the best readability of the text, is provided at the end of § 4, where the magnified areas of the thermograms previously shown in Fig. 7 and 8, are discussed in detail. In this part of the article, it is also analyzed how information on the depth of the text can be obtained from the sequence of thermograms.

These aspects have been now more stressed in the manuscript. The qualitative analysis of Fig.1 concerning the dependence of the text readability from its depth is now better introduced (1), the paragraphs were the quantitative analysis of the readability depth-dependence is carried out (2) and were the way to record and process the presented thermograms is described (3), are recalled in the text. 

(1) page 3, lines 120-130: “PT has been successfully used for the study of the historical bookbindings [21-22] with written or printed scraps inserted between the cover and the endleaf [18-19]. In this respect, PT has proved to be very effective in revealing the texts laying on such scraps that otherwise could no longer be accessed [10]. In addition, many studies carried on historical artefacts have shown that the thermographic images can even allow for their thermal reading, this result being of outmost importance to codicologists and paleographers. These studies have clearly shown how the possibility of obtaining images of the hidden text that are so sharp to allow it to be read depends on structural characteristics that, so far, have never been studied quantitatively on historical artefacts. In particular, the nature of the overlying materials, the interface between these and the text support and, above all, the depth of the text itself from the outer surface.”

(2) page 3, line 139: “In fact, as discussed more in detail later on in §4, …”

(3) page 3, lines 131-132: “Fig. 1 shows two thermograms extracted from sequences recorded and processed in the way described later on in §3.2.”

  1. Page 6: Specify the criteria for choosing the particular flash lamps and IR camera.

Two short paragraphs have been added in §3.2 to justify the choice of the adopted flash lamps e IR camera.

  • 3.2, page 8, lines 350-353: “The used flash lamps allow the light power to be adjusted to achieve an adequate signal-to-noise ratio without overexposing the artefact to excessive amounts of radiation. In addition, their light pulse was short enough to allow the applicability of the adopted theoretical model to the analysis of the experimental data.”
  • 3.2, page 8, lines 360-364: “Thanks to the above specifications, the IR camera employed provides sufficient spatial resolution and frame rate to produce spatial and time curves with an adequate amount of experimental data and a sensitivity in the operating spectral range large enough to ensure an adequate signal-to-noise ratio.”

  1. Page 7-10: Highlighting the key findings with bullet points or a summary table. The conclusion is too brief and fails to adequately summarize the key findings and their implications. It should also address potential future research directions and applications, which are currently missing.

The key findings are now summarized more effectively at the end of the Results and Discussion section as well as in the Conclusions section.

  1. There have been several papers to describe the application of infrared (IR) pulsed thermography on cultural heritage, for example:  Mercuri, F., Orazi, N., Paoloni, S., Cicero, C., & Zammit, U. (2017). Pulsed thermography applied to the study of cultural heritage. Applied Sciences7(10), 1010. Could you please illustrate the research innovation compared to them.

The research innovation the papers in the literature has been highlighted in the text and in particular at page 3, lines 111-113 and page 3, lines 117-121 and 126-130.

Round 2

Reviewer 2 Report

Comments and Suggestions for Authors

The concerns have been addressed by the authors. The paper is suggested to be published in this journal. 

Author Response

Thank you